# Testosterone Replacement Therapy in Chronic Kidney Disease Patients

**DOI:** 10.3390/nu14163444

**Published:** 2022-08-22

**Authors:** Ryszard Skiba, Aleksandra Rymarz, Anna Matyjek, Jolanta Dymus, Agnieszka Woźniak-Kosek, Tomasz Syryło, Henryk Zieliński, Stanisław Niemczyk

**Affiliations:** 1Department of General, Functional and Oncological Urology, Military Institute of Medicine, 04-141 Warsaw, Poland; 2Department of Internal Diseases, Nephrology and Dialysis, Military Institute of Medicine, 04-141 Warsaw, Poland; 3Laboratory Diagnostics Facility, Military Institute of Medicine, 04-141 Warsaw, Poland

**Keywords:** testosterone replacement therapy, chronic kidney disease, hemodialysis, serum testosterone level, erectile disfunction, total testosterone, free testosterone

## Abstract

(Background) The aim of our study was to evaluate the efficacy and safety of testosterone replacement therapy (TRT) in men with chronic kidney disease and hypogonadism on conservative and hemodialysis treatment. (Methods) The studied population consisted of 38 men on hemodialysis (HD), 46 men with CKD stages II-IV (predialysis group, PreD) and 35 men without kidney disease who were similar in age to others (control group). Serum total testosterone level (TT) was measured, and free testosterone level (fT) was calculated. Hypogonadism criteria according to the EAU definition were fulfilled by 26 men on HD (68.4%) and by 24 men from the PreD group (52%). Testosterone replacement therapy (TRT) with testosterone enanthate in intramuscular injections every 3 weeks was applied in 15 men from HD and in 14 men from PreD. The safety of TRT was monitored by measuring PSA and overhydration. (Results) A significant rise of TT and fT was observed after 3 months of TRT, but no significant changes were observed after 6 and 12 months in the HD and PreD group. An intensity of clinical symptoms of hypogonadism measured by ADAM (androgen deficiency in the ageing male) questionnaire gradually decreased, and the intensity of erectile dysfunction measured by the IIEF-5 (international index of erectile functioning) questionnaire also decreased after 3, 6 and 12 months of TRT in the HD and PreD group. (Conclusions) The applied model of TRT is effective in the correction of clinical signs of hypogonadism without a significant risk of overhydration or PSA changes.

## 1. Introduction

Chronic kidney disease (CKD) is associated with the dysfunction of multiple organs. Well recognised complications of CKD are hypertension, malnutrition, impaired glucose tolerance and insulin resistance, endocrinological disorders, electrolytes disturbances, anaemia and bone metabolism disturbances [1]. One of the endocrinological disorders which develop in association with CKD is hypogonadism [2]. It is defined as a multifactorial complex of signs and symptoms in the field of body composition, sexual and mental health and metabolic disorders which include lipid and carbohydrate abnormalities [3]. Hypogonadism is diagnosed when biochemical and clinical conditions defined by EAU are fulfilled [3]. Due to the direct impact of CKD on hypogonadism pathophysiology [4], the incidence of hypogonadism coexisting with CKD is higher than in general population and its severity is associated with the stage of CKD [5]. The incidence of hypogonadism in CKD patients stands at 27 to 66% [2]. Hypogonadism itself causes homeostasis disorders such as obesity, metabolic syndrome, anaemia, decrease of libido and erectile dysfunction, and a decrease of lean tissue mass and muscle strength [5,6,7,8,9]. Most of the aforementioned disorders coincide with hypogonadism and CKD symptoms and complications [10,11,12]. Despite a relatively high incidence of hypogonadism in CKD patients, there exist only sparse publications describing the efficacy and safety of testosterone replacement therapy (TRT) in the case of hypogonadism and CKD coexistence, which is a method of choice in hypogonadism treatment [13].

The aim of this study was to investigate the safety and efficacy of TRT in men with CKD and hypogonadism coincidence. The efficacy of TRT was estimated as an influence on hormonal disorders, body composition and erectile dysfunction. The safety of the therapy was based on the measurement of overhydration and prostate oncological evaluation.

## 2. Materials and Methods

### 2.1. Study Population

The study population consisted of 119 men aged 29–72 years. The study population was divided into three groups: hemodialysis (HD), predialysis (PreD) and control (C). There were 38 men on hemodialysis therapy, treated three times a week for more than three months in the HD group. In the PreD group, there were 46 participants with CKD stage from II to IV of CKD. Finally, C group consisted of 35 men without CKD, similar in terms of age to other men.

Exclusion criteria were as follows: age below 18 or above 75 years, lack of informed consent, lower extremity amputation, cardiac pacemaker implantation or orthopedic surgery with metal prothesis implantation, contraindications to TRT defined by EAU such as prostate cancer, breast cancer, active desire to have children, hematocrit ≥54%, uncontrolled or poorly controlled congestive heart failure, overhydration over 5 kg measured in bioimpedance spectroscopy.

All participants signed their informed consent. The study protocol was accepted by the Military Institute of Medicine Bioethics Committee; the number of approval is 107/WIM/2018. The study was performed with respect to the Declaration of Helsinki.

### 2.2. Laboratory Parameters

Among laboratory parameters, total testosterone (TT), SHBG (sex hormone binding globuline), LH(luteinizing hormone), prolactin (PRL), serum creatinine, albumin and PSA levels were measured. Free testosterone (fT) serum concentration was calculated with the use of an equation whose formula consists of total testosterone, sex hormone binding globulin (SHBG), albumin level and constant for albumin and SHBG testosterone binding available at http://www.issam.ch/freetesto.htm (accessedon 1 July 2022).

Blood samples were taken after an overnight fast in hemodialysis patients (HD group) before a mid-week dialysis session. Plasma was separated within 15 min. Plasma for assessing the SHBG level was frozen at −80 st C.

In our study we used the immunochemical method of total testosterone (TT) detection. In detail, we used the electrochemiluminescence kit Elecsys Testosterone II provided by Roche Diagnostic GmbH, Switzerland.

Serum creatinine was measured via the Jaffe method (Gen.2, Roche Diagnostics GmbH, Risch-Rotkreuz, Switzerland) and serum albumin by the colorimetric method (Gen.2, Alb 2 Roche Diagnostics GmbH, Risch-Rotkreuz, Switzerland). The following laboratory tests were performed via the electrochemoluminescency method: luteinizing hormone, PSA, PRL. The kits used were: Elecsys LH II, Elecsys total PSA and Elecsys Prolactin II, respectively. All of the above-mentioned kits were produced by Roche Diagnostics GmbH, Switzerland. Serum SHBG was measured via the immunoenzymatic sandwich test, produced by LDN.

CKD stages was ascertained according to Kidney Disease Improving Global Outcomes (KDIGO) based on glomerular filtration rate (GFR) estimated using short Modification of Diet in Renal Disease (MDRD) formula.

Hypogonadism was defined according to EAU definition (total testosterone <8 mmol/L or total testosterone in range of 8–12 mmol/L and free testosterone below 220 pmol/L). Besides, laboratory parameters participants were evaluated in a field of clinical sings of hypogonadism. Erectile dysfunction (ED) was measured by the IIEF-5 (international index of erectile functioning) questionnaire. To quantify the severity of hypogonadism, we used the ADAM (androgen deficiency in the ageing male) questionnaire. Moreover, participants underwent a physical examination to reveal hypogonadism signs.

### 2.3. Body Composition

Bioimpedance spectroscopy was performed using the Body Composition Monitor, Fresenius Medical Care. An examination was done in supine recumbency and electrodes were placed in tetrapolar configuration. In hemodialysis patients, an examination was performed before dialysis session. In patients with arterio-venous fistula, electrodes were placed on the upper arm opposite the vascular access.

### 2.4. Testosterone Replacement Therapy

After an initial evaluation of the HD and PreD group, we selected the hypogonadal group with respect to criteria of hypogonadism defined by EAU. The hypogonadal group received TRT with testosterone enanthate injected intramuscular in 3-week intervals. During the therapy, we performed evaluation of its efficacy and safety in a period of 3, 6 and 12 months by repeating the aforementioned tests.

## 3. Results

119 men were included in the study. The study groups did not differ significantly in terms of age (*p* = 0.486). Total and free median testosterone levels were the lowest in HD group (TT 2.8 ng/mL, fT 44.5 pg/mL) slightly higher in PreD group (TT 3.4 ng/mL, fT 60.4 pg/mL) and the highest in C group (TT 4.17 ng/mL, ft 71.9 pg/mL) and the values differed significantly (*p* < 0.0001). LH and PRL median serum levels were the highest in HD group (LH 11,1 IU/L, PRL 21.7 ng/m) and values also differed significantly (*p* < 0.0001). With regard to EAU hypogonadism definition, we have verified symptoms of hypogonadism with the ADAM questionnaire and compared it with laboratory results. This evaluation revealed hypogonadism in 68.4% participants in the HD group, 52% participants in the PreD group and only 25.7% participant in the C group. The differences were statistically significant (*p* = 0.001). Serum albumin levels differed significantly between the groups (*p* = 0.0003) with the lowest value in HD group.

We observed statistically significant differences among study groups in terms of frequency of ED measured by the IIEF-5 questionnaire. The highest frequency presented was the PreD group (82.6%), slightly lower was the HD group (78.9%) and the lowest was the C group (40%).

Considering body composition parameters, lean tissue mass represented by average LTI was lowest in the HD group (13.1 kg/m^2^) and the values were significantly different (*p* = 0.003). With respect to BMI, it was predictable that the highest FTI was noticed in PreD group (15.3 kg/m^2^), values also was significantly different among study groups (*p* = 0.031). Overhydration was statistically higher in HD group than in PreD and C group (2.6 L vs 0.3 L and 0.6 L respectively *p* < 0.0001). Initial biochemical, anthropometric and body composition data are presented in Table 1.

After initial evaluation, we have selected candidates to TRT from the study group with respect to EAU hypogonadism definition and study exclusion criteria. Finally, TRT was applied in 15 men from HD and in 14 men from PreD.

As a result of TRT, we observed a significant rise of TT and fT serum concentrations in HD and PreD group (Figure 1 and Figure 2). A significant difference was observed after 3 months of follow-up and further observations, after 6 and 12 months, revealing no significant changes.

Median LH serum concentrations gradually decreased after 3, 6 and 12 months of observation in HD group in significant manner *p* = 0.0002. A similar phenomenon was observed in PreD group (*p* = 0.0003) (Table 2).

In the HD group during TRT, PRL levels increased after 3 months of observation and decreased after 6 and 12 months in comparison to the levels after 3 months. In the PreD group PRL serum concentrations raised systematically and significantly after 3, 6 and 12 months (*p* < 0.05) (Table 2).

The intensity of clinical symptoms of hypogonadism measured by the ADAM questionnaire gradually decreased during TRT after 3, 6 and 12 months in HD and PreD group. In HD group initially median ADAM score was 7 IQR (5–8) points and after 12 months of TRT it decreased to 3 (IQR 2–4) points (*p* < 0.0001). In the PreD group, initial median ADAM score was 7 (IQR 5–8) points and after 12 months of treatment decreased to 3 (IQR 1–4) points (*p* < 0.0001) (Figure 3).

TRT was associated with the decrease of intensity of ED measured by IIEF-5 questionnaire in both HD and PreD group. Initial median IIEF-5 score in the HD group was 4 (IQR 2–12) points. After 12 months of TRT it raised to 16 (IQR 12–19) points and the change was statistically significant (*p* < 0.0001). Similar change was observed in the PreD group where the initial IIEF-5 median score was 13 (IQR 6–18) points and after a year of observation it increased to 17 (IQR 13–19) points (*p* = 0.004) (Figure 4).

Among body composition parameters we observed a decline of mean LTI from initial 13.9 ± 1 kg/m^2^ to 12.7 ± 0.8 (*p* = 0.003) after 12 months in HD group. In the same group mean FTI increased after year of observation from initial 14.4 ± 2.9 kg/m^2^ to 17.4 ± 3.2 (*p* = 0.001). Corresponding parameters observed in the PreD group did not change in a study duration under the influence of TRT. Overhydration in the HD group did not change significantly (*p* = 0.06) during observation. In PreD group overhydration increased significantly from the mean value −0.4 to 1.2 L (*p* = 0.004) during one year of observation. The reference range to overhydration measured by BCM is from −1 to 1 L (Table 2).

PSA values during one year of TRT did not change significantly in both study groups. Similar findings were noted in albumin concentration with borderline significance of change in Pred group (Table 2).

## 4. Discussion

The diagnosis of hypogonadism in the population of men suffering from CKD worsens course of comorbidities and is associated with increased morbidity and mortality [14]. Hypogonadism and CKD are connected by self-dependent pathophysiological processes, and the risk of hypogonadism increases along with CKD progression expressed by GFR decline. In our study we observed that hypogonadism is more frequent in advanced stages of CKD reaching the level of 68.4% in HD group. To compare the percentage of hypogonadal men in PreD group was 52% and in C group was 25.7%. These observations correspond with other studies. In the study by Carrero et al. the frequency of hypogonadism varied from 27 to 66% depending on the stage of CKD [2].

Aside from testosterone deficiency, a clinical image of hypogonadism includes other endocrinological disorders such as hyperprolactinemia [15]. In our study, the intensity of hyperprolactinemia was more expressed in the HD group (median PRL serum concentration 21.7 ng/mL) than in the PreD group (median PRL serum concentration 10.1 ng/mL *p* < 0.0001). The observation mentioned above may be confirmed by another study published by Carreo et al. where the frequency of hyperprolactinemia is dependent on stage of CKD and varies from 30% in early stages to 66% in patients undergoing hemodialysis [16]. Luteinizing hormone concentration regarding the course of CKD remains high but amplitude of secretion is decreased, and pulsating character of secretion is spared [5]. In our study initially median LH serum concentration was 11.1 IU/L for HD group and 8.5 IU/L for PreD group.

Erectile dysfunction in the population of men with CKD has a multifactorial etiology including endocrinological disorders (hypogonadism, hyperprolactinemia, secondary hyperparathyroidism), cardiovascular disorders, peripheral neuropathy, anemia, depression, obesity and side effects of medications. The frequency and intensity of ED increase along with the severity of CKD. More advanced stages of CKD are associated with higher incidence and intensity of ED which was described in some studies. Bellinghieri et al. noticed positive correlation between score in IIEF questionnaire and eGFR in patients with CKD on conservative treatment [17]. In our study frequency of ED was 78.9% in HD group and 82.6% for PreD group. Slightly higher frequency of ED disfunction in less severe CKD can be associated with the fact that in PreD group had higher BMI (31 kg/m^2^) than HD group. However, the intensity of ED was significantly more severe in HD group. The Median IIEF-5 score before TRT was 4 points for HD group in comparison to 13 points for PreD group.

Considering body composition disorders in a course of CKD one should stress its impact on increase of morbidity and mortality in CKD population [18]. One of the criteria of definition of protein energy wasting (PEW) is loss of lean tissue mass which is testosterone dependent [19] and that relation is bidirectional. The initial evaluation of our study population revealed statistically significant difference between mean LTI in three studied groups (*p* = 0.045). The lowest LTI was observed in advanced stages of CKD represented by men treated with HD. Above mentioned observation can be confirmed be study published by Nixon et al. [20].

Regarding body composition disorders associated with fat tissue mass we observed in our study highest fat tissue mass represented by FTI in PreD group. Such an observation seems to be slightly different than those published by Cobo et al. where authors noticed reverse correlation between serum testosterone concentration in CKD stages II-IV group and fat tissue mass. These phenomena can be explained by higher mean body mass in PreD group and PEW which intensity is associated with hemodialysis and connected with general body mass loss including fat tissue mass loss [21].

TRT in this study was performed with testosterone enanthate (TE). It is a short acting agent, which activity lasts from 2 to 4 weeks. The drug selection was made regarding to EAU guidelines which suggest using short acting agents on the beginning of TRT to enable physician a quick interruption of TRT in case of side effects occurrence. Moreover, we decided to continue TRT with TE because of the study group profile, which is more vulnerable on risk of overhydration, one of the possible side effects of TRT. TE characteristics reveal a tendency to produce fluctuating serum testosterone levels with rapid raise after injection to supraphysiological levels and rapid decrease early after injection [22]. In our study we observed significant improvement of serum testosterone levels (Figure 1 and Figure 2). In supplemented men treated with HD we observed significant increase of mean TT serum concentration form 2.56 ng/dL to 7.45 ng/mL after 3 months of therapy. Further observation after 6 and 12 months of TRT reveals no statistically significant changes. Similar changes were observed in free testosterone levels. Mean serum concentration of fT in HD group increased from 43.58 pg/dL to 154.9 pg/dL after 3 months of TRT. Further observations reveal no statistically significant changes. In PreD group receiving TRT the initial TT concentration increased from 2.7 ng/mL to 4.8 ng/dL after 3 months. Free testosterone level increased from 49 pg/dL to 99.6 pg/dL after 3 months of therapy. The observations after 6 and 12 months revealed no statistically significant changes both for TT and fT serum concentrations.

Despite the satisfactory results achieved with TRT on TT and fT concentrations we observed relatively high concentration of LH after 3 months of TRT, fluctuating on upper laboratory limit. In HD group median LH was 8.01 IU/L and in PreD group 6.24 IU/L. The above-mentioned facts and mechanism of feedback between testosterone and LH serum concentration suggests that despite the satisfactory impact of TRT on testosterone levels at checkpoints its concentration between them might be lower.

In our study we observed unexpected changes in PRL concentration during observation. In HD group we observed a rise of PRL concentration after 3 months of TRT with further decline after 6 and 12 months. In PreD group PRL concentrations raised during whole period of observation. The explanation of this phenomenon can be associated with the fact that hyperprolactinemia is associated with CKD progression and observed changes were independent from TRT. Another reason for observed findings can be an aromatization of exogenous testosterone to estradiol which stimulates pituitary gland to PRL production and cause secondary hyperprolactinemia. This phenomenon was described by Sodi et al. who also noticed hyperprolactinemia after TRT and proposed this explanation [23].

Observation of clinical symptoms of hypogonadism measured by ADAM reveals gradually decline of its intensity in HD and PreD group (*p* < 0.0001). In HD group we observed a reduction of ADAM score from initial 7 to 3 points after 12 months of TRT. In PreD group we observed reduction of ADAM score from initial 7 to 3 pts after 12 months of TRT. Study made by Yamaguchi et al. in population without kidney disease revealed reduction of ADAM score after 6 months of TRT with TE from initial mean 8.3 to 6.8 points. On the other hand, Yeo et al. published study on TRT in CKD population and observed a decline Ageing Male Symptoms Scale score from 48.72 to 33.11 points after 3 months of TRT with testosterone transdermal system [24]. Our results are more similar to effects achieved by Yamaguchi et al. rather than to those described by Yeo et al. The explanation may be associated with different testosterone agents used by Yeo et al. which provide more stable testosterone serum concentration during therapy.

TRT in our study resulted in reduction of ED severity in both HD and PreD group. In HD group initial median IIEF-5 score was 4 and after 12 months of TRT rose to 16 points. In PreD group initial IIEF-5 score was 13 and after one year of TRT raised to 17. Meta-analysis prepared by Corona et al. reported mean improvement in IIEF-5 score by 2.31 points. Moreover, authors observed higher efficacy of TRT on ED in men with TT bellow 2.31 ng/mL but with connection to phosphodiesterase-5 inhibitors [25]. Study prepared by Cangüven et al. presented efficacy of transdermal TRT in ED treatment in hemodialysis patients. In this study, the authors achieved an enhancement in IIEF-5 score without total resolve of ED [26]. Results of our study and literature analysis indicate that TRT in ED treatment should be considered as a supplementary method.

Regarding body composition disorders during TRT in our study, we observed significant decrease of lean tissue mass expressed by LTI (*p* = 0.003) and increase of fat tissue mass expressed by FTI (*p* = 0.001) in HD group. In the PreD group these parameters did not change significantly. Results of the studies made on general population indicated that TRT provided increase of lean tissue mass and muscle strength [27]. Conclusions from Testosterone Trials revealed an enhancement effect of TRT on effort tolerance but without statistical significance [28]. The influence of androgen therapy on body composition in CKD population was tested in a few studies. One of them was prepared by Johansen et al. [29]. Authors used weekly injections of 100 mg of nandrolone which caused an increase of lean tissue mass and muscle strength in hemodialysis patients. However, nandrolone is a synthetic androgen which anabolic potential is 11 times higher than observed for testosterone [30]. A clinical trial which tested effect of transdermal TRT in CKD population on body composition was presented by Yeo et al. [24]. In this study, the authors reported an increase of handgrip strength without significant BMI reduction. The reason why results achieved in our study was different than presented in the literature is probably diverse therapeutic potential of used androgen agent and catabolic effect of PEW phenomenon associated with CKD which overbalanced anabolic potential of TRT.

In our study we performed an evaluation of TRT safety including risk of overhydration and prostate oncologic disorders. In the study groups TRT had to be ceased in one patient due to the increase of PSA serum concentration. Digital rectal examination in this case was normal. Regarding PSA progression, mpMRI examination was performed and scan reveals no risk of clinically significant prostate cancer. TRT cessation results in decrease of PSA serum concentration to normal value. In the rest of the study group, we did not observe significant changes in serum PSA levels. Meta-analysis made by Bolye et al. revealed no significant PSA serum concentration increase in one year observation, which confirms our study results [31]. Overhydration during the one-year observation in the HD group did not change significantly. In the PreD group changes were statistically significant but not clinically significant. An increase of overhydration from −0.4L to 1.4 L in patients with renal failure and with normal daily diuresis had no negative impact on the clinical course of the patients and can be even favorable. Moreover, reference range for overhydration in bioimpedance spectroscopy performed by BCM is from −1 L to 1 L. In the study published by Lawrence et al. authors observed fluid retention in 4 out of 27 participants, but the phenomenon was not clearly connected with TRT. To conclude, we did not observe a clinically significant increase of fluid retention during TRT in patients with CKD; however, hydration status during this therapy should be screened.

## 5. Conclusions

In our study, TRT effectively corrected endocrinological disorders associated with hypogonadism associated with CKD; however, testosterone enanthate may produce unstable serum concentration. The applied model of TRT is effective in the correction of clinical signs of hypogonadism and ED. The influence of TRT on body composition disorders is questionable; however, it requires further study with other testosterone formulations.

## Figures and Tables

**Figure 1 nutrients-14-03444-f001:**
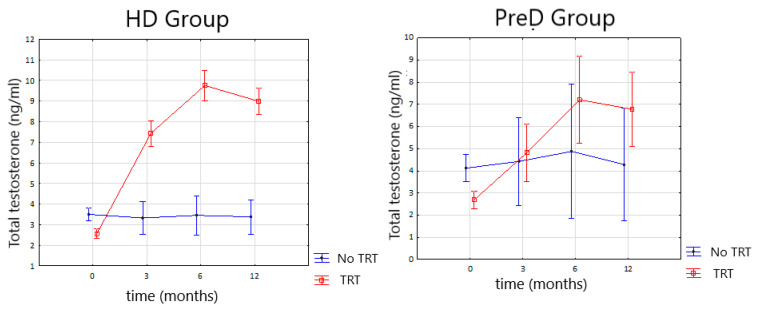
Changes in the total testosterone serum concentration during TRT (testosterone replacement therapy) in the HD (hemodialysis) and PreD (predialysis) groups.

**Figure 2 nutrients-14-03444-f002:**
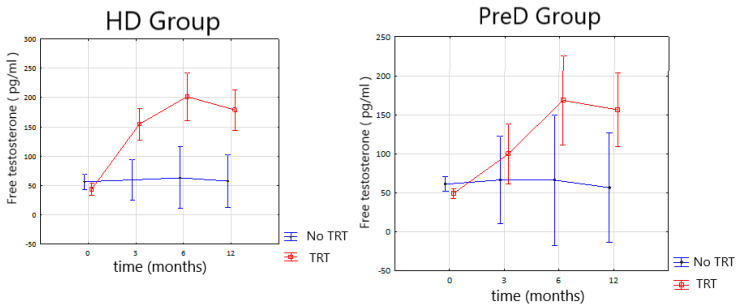
Changes in the free testosterone serum concentration during TRT (testosterone replacement therapy) in HD (hemodialysis) and PreD (predialysis) group.

**Figure 3 nutrients-14-03444-f003:**
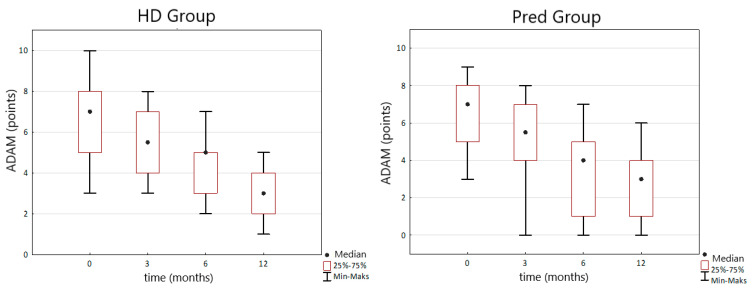
Changes in the ADAM (androgen deficiency in the ageing male) score during TRT (testosterone replacement therapy) in the HD and PreD groups.

**Figure 4 nutrients-14-03444-f004:**
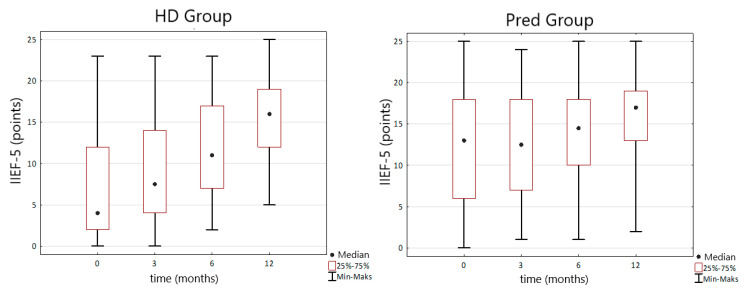
Changes in the IIEF-5 (international index of erectile functioning) score during TRT (testosterone replacement therapy) in the HD and PreD groups.

**Table 1 nutrients-14-03444-t001:** The initial group characteristics.

Parameter	HD Group *n* = 38	Pred Group *n* = 44	Control Group *n* = 35	*p*
Demographic data				
Age (years)	58 (50–65)	64 (53–67)	59 (47–69)	0.486
Diabetes				
Yes No No data	18 (47.4%) 18 (47.4%) 2 (5.2%)	19 (41.3%) 24 (52.2%) 3 (6.5%)	6 (17.1%) 28 (80%) 1 (2.9%)	**0.038**
Anthropometric measures				
Weight (kg)	89 (72–104)	96 (87–110)	82 (75–93)	**0.003**
Height (cm)	175 (171–178)	175 (172–178)	173 (166–178)	0.155
BMI (kg/m^2^)	29 (24–34)	31 (28–35)	27 (26–310)	**0.012**
Hormonal status				
TT (ng/mL)	2.8 (2.3–3.7)	30.4 (2.7–4.7)	4.17 (2.8–5.6)	**0.001**
fT (pg/mL)	44.5 (30.6–62.6)	60.4 (45.1–66.2)	71.9 (58.2–81.1)	**<0.0001**
SHBG (nmol/L)	45.9 (33.6–58.3)	41.8 (32.9–54.4)	44.9 (27.2–57.1)	0.698
PRL (ng/mL)	21.7 (17.2–40.3)	10.1 (8.4–13.2)	10.1 (8–12.6)	**<0.0001**
LH (IU/L)	11.1 (8.7–15.3)	8.5 (6.9–11.8)	5.3 (3.6–6.5)	**<0.0001**
Biochemical measurement				
Serum creatinine (mg/dL)	8 (6.3–10.5)	3.4 (2.7–4.7)	1 (0.8–1.1)	**0.001**
Albumin (g/dL)	4.15 (3.9–4.4)	4.5 (4.2–4.8)	4.5 (4.2–4.6)	**0.0003**
PSA (ng/mL)	0.8 (0.5–1.6)	1.2 (0.8–2.1)	1.2 (0.8–2)	0.926
Bioimpedance spectroscopy				
LTI (kg/m^2^)	13.1 (2.9)	15.3 (2.5)	16.1 (3)	**0.003**
FTI (kg/m^2^)	14.3 (7.4)	15.6 (5.4)	11.8 (5.1)	**0.031**
TBW (L)	42.7 (6.8)	44 (5.8)	44 (7.8)	0.077
ECW (L)	21.3 (3.6)	20.7 (2.8)	19.5 (2.1)	**0.002**
ICW(L)	21.4 (3.8)	23.7 (3.6)	23.5 (4.4)	**0.005**
BCM (kg)	23.2 (6.5)	26.6 (6.1)	28.2 (8)	**0.007**
OH (L)	2.6 (1.7–4.5)	0.3 (−0.6–1.5)	0.6 (−0.3–1)	**<0.0001**
Erectile dysfunction				
Yes (IIEF ≤ 21) No (IIEF > 21) No data	30 (78.9%) 5 (13.2%) 3 (7.9%)	38 (82.6%) 8 (17.4%) 0	14 (40%) 3 (8.6%) 18 (51.4%)	**<0.0001**
Hypogonadism				
Yes No	26 (68.4%) 12 (31.6%)	24 (52%) 22 (48%)	9 (25.7%) 26 (74.3%)	**0.001**

Bolded values are statistically significant. BCM, body cell mass; BMI, body mass index; ECW, extracellular water; fT, free testosterone; FTI, fat tissue index; HDL, high density lipoprotein; ICW, intracellular water; IIEF-5, international index of erectile functioning, LDL, low density lipoprotein; LH, luteinizing hormone; LTM, lean tissue mass; LTI, lean tissue index; OH, overhydration; PSA, prostate specific antigen; PRL, prolactin; SHBG, sex hormone binding globulin; TBW, total body water; TT, total testosterone.

**Table 2 nutrients-14-03444-t002:** Changes of the observed parameters during TRT.

Parameter	Checkpoint	HD Group *n* = 15	*p*	Checkpoint	Pred Group *n* = 14	*p*
Laboratory measurements						
Serum creatinine (mg/dL)	0	7.9 (6.1–11.7)	0.226	0	1.9 (1.6–3.5)	0.945
	3	10 (7.6–10.7)		3	2 (1.5–3)	
	6	9.5 (7.7–10.5)		6	1.8 (1.6–3.3)	
	12	9.6 (8.2–10.3)		12	2.1 (1.5–3.6)	
Albumin (g/dL)	0	4.2 (0.1)	0.131	0	4.6 (0.1)	**0.01**
	3	4.1 (0.1)		3	4.5 (0.1)	
	6	4.1 (0.1)		6	4.4 (0.1)	
	12	4.1 (0.1)		12	4.3 (0.1)	
PSA (mg/dL)	0	0.9 (0.4–1.1)	0.271	0	1 (0.8–1.9)	0.079
	3	0.6 (0.5–0.9)		3	1.2 (1–1.6)	
	6	0.6 (0.4–1.2)		6	1.2 (0.9–1.4)	
	12	0.8 (0.5–1.2)		12	1.2 (1.1–1.6)	
PRL (ng/mL)	0	21.7 (17.2–45.4)	**0.01**	0	10.4 (8.2–13.1)	**0.003**
	3	43.1 (21.4–87.1)		3	13.3 (8.9–16.8)	
	6	30 (17.1–75.5)		6	15.3 (13–19.2)	
	12	27.1 (17.2–72)		12	17.1 (13.7–21)	
LH (IU/L)	0	8.7 (8–14.3)		0	7.8 (6.9–8.9)	**0.0003**
	3	8 (0.8–18.6)		3	5.3 (3.1–8.4)	
	6	3.5 (1.2–9.7)		6	0.9 (0.6–3.9)	
	12	2.6 (1.6–7.6)		12	1.8 (0.9–3.3)	
Bioimpedance spectroscopy						
LTI (kg/m^2^)	0	13.9 (1)	**0.003**	0	15.6 (0.8)	0.64
	3	15.1 (1)		3	16.6 (1.2)	
	6	12 (0.7)		6	15.3 (1.4)	
	12	12.7 (0.8)		12	14 (0.8)	
FTI (kg/m^2^)	0	14.6 (2.9)	**0.001**	0	18,1 (1.5)	0.08
	3	14.3 (3)		3	16,8 (1.9)	
	6	17.4 (3.1)		6	18,4 (1.9)	
	12	17.4 (3.2)		12	19,6 (1.4)	
BCM (kg)	0	23.1	**0.003**	0	27.1 (2)	0.196
	3	25.6		3	29.4 (2.5)	
	6	19.2		6	26.9 (3.1)	
	12	20.4		12	23.9 (1.7)	
OH (L)	0	3.4 (0.7)	0.06	0	−0.4 (0.4)	**0.004**
	3	2.4 (0.9)		3	0 (0.5)	
	6	2.1 (0.9)		6	−0.4(0.6)	
	12	1.9 (0.7)		12	1.2 (0.5)	

Bolded values are statistically significant. BCM, body cell mass; BMI, body mass index; ECW, extracellular water; FTI, fat tissue index; HDL, high density lipoprotein; ICW, intracellular water; IIEF-5, international index of erectile functioning, LDL, low density lipoprotein; LH, luteinizing hormone; LTM, lean tissue mass; LTI, lean tissue index; OH, overhydration; PSA, prostate specific antigen; PRL, prolactin; TBW, total body water.

## Data Availability

Not applicable.

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
