# Peer review of "Testosterone Replacement Therapy in Chronic Kidney Disease Patients"

_nutrients, 2022, doi:10.3390/nu14163444_

Round 1

Reviewer 1 Report

1. Although the investigators described how creatinine measured, the authors have not described how CKD was ascertained? CKD and staging was identified by CKD-MDRD? CKD-EPI? ICD codes?

2. Figure and Table legends, please provide all abbreviations. 

3. Analyses based on CKD staging by subgroups are encouraged if possible.

Author Response

Dear reviewer thank You for your opinion, we have prepared answers for Your comments. 

1) In section materials and methods we have inluded information about methodology of used CKD staging and GFR calculating.

2) Missing abbreviations explanation were included.

3) Altough analisis based on subgroups formed by CKD stages will provide further interesting information and probably will develop more interesting conlusions we have not performed such an analysis because it will effect in further study group fragmentation. Divison relatively small study group in smaller subgroups will effect in difficulties in statisticall analysis which may efect in inconlusive resoults. 

Reviewer 2 Report

The study evaluates the efficacy and safety of testosterone replacement therapy in men with chronic kidney disease and hypogonadism. I recommend the paper be published with minor revisions.

1. The authors should cite the publication :Testosterone replacement therapy in hypogonadal men: assessing benefits, risks, and best practices; DOI: 10.3810/pgm.2008.09.1914  in the introduction. 

2. Throughout the draft, the numbers have a problem of having commas instead of a decimal. For eg. Line 112, TT 2,8 ng/ml , fT 44,5 pg/ml should be 2.8 ng/ml, fT 44.5 pg/ml and so on. 

Author Response

Dear reviewer thank You for Your suggestions. 

We have prepared manuscript corections following Your suggestions. 

1) Sugested citation was included. 

2) Commas have been changed into decimal. 
